# Calibration of a Stereoscopic Vision System in the Presence of Errors in Pitch Angle

**DOI:** 10.3390/s23010212

**Published:** 2022-12-25

**Authors:** Jonatán Felipe, Marta Sigut, Leopoldo Acosta

**Affiliations:** 1Instituto Tecnológico y de Energías Renovables, 38600 Granadilla, Spain; 2Departamento de Ingeniería Informática y de Sistemas, Facultad de Ciencias, Universidad de La Laguna, 38200 San Cristóbal de La Laguna, Spain

**Keywords:** calibration, machine learning, U-V disparity, 3D scenes reconstruction

## Abstract

This paper proposes a novel method for the calibration of a stereo camera system used to reconstruct 3D scenes. An error in the pitch angle of the cameras causes the reconstructed scene to exhibit some distortion with respect to the real scene. To do the calibration procedure, whose purpose is to eliminate or at least minimize said distortion, machine learning techniques have been used, and more specifically, regression algorithms. These algorithms are trained with a large number of vectors of input features with their respective outputs, since, in view of the application of the procedure proposed, it is important that the training set be sufficiently representative of the variety that can occur in a real scene, which includes the different orientations that the pitch angle can take, the error in said angle and the effect that all this has on the reconstruction process. The most efficient regression algorithms for estimating the error in the pitch angle are derived from decision trees and certain neural network configurations. Once estimated, the error can be corrected, thus making the reconstructed scene appear more like the real one. Although the authors base their method on U-V disparity and employ this same technique to completely reconstruct the 3D scene, one of the most interesting features of the method proposed is that it can be applied regardless of the technique used to carry out said reconstruction.

## 1. Introduction

In [1], the authors propose a method for the complete reconstruction of 3D scenes based on U-V disparity and conduct an analytical study of its behavior and sensitivity to errors in the pitch angle, or θ angle, of the stereoscopic vision system. The pitch angle (θ) is the angle between the optical center and the horizontal plane. Errors in θ cause the calculated planes to deviate from the ideal planes, which are the planes that would exist in the absence of errors, by a certain amount that is measured as the difference between the normal of the calculated and ideal planes. The study carried out allows us to conclude that this deviation, which generally depends on the orientation of the plane considered, is always less than or equal to the error in the pitch angle. In addition, the probability distribution is not uniform, since although in some cases the deviation of the plane is zero or very small, the probability that a plane of the scene deviates from the ideal by the greatest amount possible, which matches the error in the pitch angle, is very high.

While the study carried out in [1] shows that an error in the pitch angle of the stereoscopic vision system produces distortion in the reconstructed scene, it also raises a question of great practical interest: can the error in the pitch angle be estimated in some way, and can this estimate be used to ‘correct’ the calculated planes such that they are as close as possible to the ideal planes? In other words, with the information available, can the stereoscopic vision system be calibrated so that the reconstructed scene is as close as possible to the real one? In this paper, we answer this question in the affirmative and propose a calibration procedure that relies on the use of artificial intelligence techniques and, more specifically, machine learning.

As is well known, calibration is a process that consists of comparing the values obtained with a given measurement instrument or technique with the corresponding measurement of a reference standard. In the literature, there are countless contributions in which different techniques are proposed for calibrating both monocular and binocular vision systems. Although we will focus on the latter, [2] contains a good example of the automatic calibration of a monocular vision system for extracting and tracking obstacle 3D data from the surrounding environment of a vehicle in traffic.

In the field of stereoscopic vision, the calibration problem is broader since there are different elements to be calibrated. For example, the cameras can be calibrated individually, in which case the approach adopted in [2] and similar papers would be valid; but the position of one camera relative to the other, and even the position of the full stereo pair, can also be calibrated. In any case, this problem has been considered and solved using a wide range of approaches. Below is a selection of some of them in the context of 3D reconstruction, which simply seeks to illustrate the wide variety of works in this field. In [3], the authors focus on studying the relationship between reconstruction accuracy and calibration accuracy, discussing the main factors involved in reconstruction errors. In [4], the problem of the propagation of input data errors in the stereovision process and its influence on the quality of reconstructed 3D points is addressed. In this case, the authors focus on camera calibration and 3D reconstruction algorithms that employ methods based on singular value decomposition (SVD). A set of original methods for assessing the accuracy of camera parameters is presented in [5]. Some of them have a critical influence on the quality of the 3D reconstruction process and thus require special attention during calibration. While the works mentioned focusing more on analyzing the influence of calibration errors on the quality of the 3D reconstruction, in other cases different methods are proposed for carrying out this calibration. For example, in [6] the authors employ a special flat board that uses circular spots of different sizes to calibrate the cameras. Based on an existing algorithm for calibrating a camera, in this work, the intrinsic and radial distortion parameters of the lens can be calculated using several pairs of images of the flat board that are captured by two cameras in different orientations. A 3D surface scanner that combines stereovision and slit-scanning is presented in [7]. The camera pair is calibrated using pairs of synchronized images from an encoded chessboard. Elsewhere, applications for autonomous navigation systems frequently resort to markings on the road to calibrate the vision system. For example, in [8] the system proposed detects pedestrian crossings and extracts their corners in the two images of the stereo pair, which are matched in a later step. The authors demonstrate that the technique proposed is capable of accurately extracting and matching the corners of pedestrian crossings, and of recalibrating the stereovision system more accurately and robustly than with other similar procedures.

Along this same line, we also find several patented automatic calibration procedures. To cite two examples, in [9] the author proposes an on-board system for motor vehicles in which the pitch and yaw errors are determined from the right and left images of the same scene, which must contain at least one lane of the road on which the vehicle is traveling. In the invention presented in [10], also for motor vehicles, the authors propose a system for accurately estimating in real time the error value in the yaw angle between the left and right cameras. According to the invention, a set of equations is solved by using a non-linear equation solver method to obtain an estimated value for the intrinsic yaw error.

Artificial Intelligence techniques and, more specifically, Machine Learning and Deep Learning techniques, have been applied for years to solve problems of very different natures, yielding satisfactory results in many cases. Numerous authors use this type of technique to calibrate different devices. For example, in [11], four different models are trained for radio map calibration in RSS-fingerprinting-based positioning systems. These four models are: fitting a log-distance path loss model, Gaussian Process Regression, Artificial Neural Network, and Random Forest Regression. In [12], machine learning is used to calibrate a physical model that tries to reproduce the vibrational behavior of an overhead line conductor, which is a novel approach to the problem in question. To achieve their goal, the authors train and test different models. The literature also contains a vast number of applications in the field of computer vision. In [13], a study of the state of the art of deep learning-based gaze estimation techniques is presented, focusing on Convolutional Neural Networks (CNN). The same work presents a review of other machine learning-based gaze estimation techniques. This study aims to empower the research community with valuable and useful insights that can enhance the design and development of improved and efficient deep learning-based eye-tracking models. In [14], the authors propose a vision-based robotic solution for wire insertion in which a priori knowledge of the scenario is used with a calibrated RGB camera and a robotic arm. The solution presented combines different techniques (gradient-based, trained classifiers, and stereovision) to obtain standard images. Another application of machine learning in the field of computer vision is presented in [15]. In this case, a solution based on deep learning is proposed to automatically control the closing and sealing of pizza wrappers. To train the networks, the authors propose a classification of the defects in the pizza wrappers that focuses on the sealing and closing, and an image-based method capable of automatically detecting them. To cite three more examples, in [16] the authors propose a convolutional neural network (CNN) trained on a large set of example chessboard images for camera calibration. In [17], multiple linear regression, support vector regression, and random forest regression are applied to calibrate low-cost air monitoring sensors in the field. The same problem is considered in [18]. In this case, the authors explore methods to additionally improve the calibration algorithms to increase the measurement accuracy by considering the impact of temperature and humidity on the readings, by using machine learning. A detailed comparative analysis of linear regression, artificial neural networks, and random forest algorithms is presented.

As concerns the use of artificial intelligence to calibrate stereoscopic vision systems, there are interesting applications in the literature. In [19], the authors propose a method based on a Back Propagation neural network optimized with an improved genetic simulated annealing algorithm (IGSAA-BP) to calibrate a binocular camera, improving on the precision and speed of convergence that is obtained with BP neural networks. In [20], a Radial Basis Function Network (RBFN) is used to provide effective methodologies for solving difficult computational problems in both camera calibration and the 3D reconstruction process.

This paper proposes a procedure that relies on machine learning techniques, and specifically on various regression algorithms, to calibrate a stereoscopic vision system by estimating the error in the pitch angle of the cameras. Once the error is estimated, it can be used to correct the deviation of the cameras. As far as the authors know, this is a novel contribution that is also independent of the specific technique used to reconstruct the scene from a pair of stereoscopic images.

## 2. Materials and Methods

U-V disparity is a way to represent the information contained in a pair of stereo images to extract depth information. It does so by building a three-dimensional space in which the vertical coordinate of a point projected in one of the images of the stereo pair, the horizontal coordinate of the same point, and the difference between the horizontal coordinates of the projection of the point in both images, are represented. Despite the fact that there is a large number of contributions regarding multiple view geometry (see, for example, [21,22]), the authors continue in the line of our previous work representing the 3D scene as a combination of planes, where each of them corresponds to an element of the scene. In [1], the authors describe in detail the technique used to extract the necessary information from this three-dimensional space and interpret it to reconstruct a 3D scene using planes. Only those concepts needed to describe the proposed calibration technique will be included in this paper.

### 2.1. 3D Reconstruction Based on Planes

Figure 1 shows the stereoscopic system considered, with a single degree of freedom corresponding to the pitch angle.

The following parameters are shown in the figure above:*d*: Distance between the optical centers $\left(O_{i},i=r,l\right)$ of both cameras.*h*: Height of the optical center of the camera above the ground.θ: The angle between the optical center and the horizontal plane (pitch angle)

The projection of points $P={\left(XYZ1\right)}^{T}$ in the global reference system ($R_{w}$) at the corresponding coordinates ($\left(u_{0},v_{0}\right)$) within the coordinate systems of each camera ($R_{l}$ and $R_{r}$) also requires defining the following variables:*f*: Focal length of the cameras, assumed equal in both cameras.$t_{u},t_{v}$: Size of the pixels in the *u* and *v* dimensions.

The relationships drawn from Figure 1 between the different projections allow us to define the Disparity Space $\left(u_{r},v,\Delta \right)$.
(1)$$\begin{array}{ccc} & & u_{r}=u_{0}+\frac{\alpha X-\alpha \frac{d}{2}}{\left(Y+h\right)\sin\theta +\left(Z+f\right)\cos\theta }\\ & & v=\frac{Y\left(\alpha \cos\theta +v_{0}\sin\theta \right)+\left(v_{0}\cos\theta -\alpha \sin\theta \right)\left(Z+f\right)}{Y\sin\theta +\left(Z+f\right)\cos\theta +h\sin\theta }+\frac{\alpha h\cos\theta +v_{0}h\sin\theta }{Y\sin\theta +\left(Z+f\right)\cos\theta +h\sin\theta }\\ & & \Delta =\frac{\alpha d}{\left(Y+h\right)\sin\theta +\left(Z+f\right)\cos\theta } \end{array}$$

As described in [1], those regions of space in which the gradient of the disparity is constant represent the projection of points that are part of the same object. This result can be used to completely characterize the content of a 3D scene in which each element in it is represented by the plane that contains the points that make up the object, defined by the general plane equation:(2)rX+sY+tZ+u=0

All that is needed is to have 3 points contained in the region. According to Equation (Equation 1), each point i∈[1,3] can be projected into the disparity space, obtaining $u_{ri}$, $v_{i}$ and $\Delta_{i}$. The coefficients that characterize the plane considered can be calculated from the information contained in the Disparity Space with the Equations (Equation 3)–(Equation 6).
(3)$$\begin{array}{cc} r & =\frac{d^{2}}{\Delta_{1}\Delta_{2}\Delta_{3}}\alpha \left(\left(v_{2}-v_{1}\right)\Delta_{3}+\left(v_{1}-v_{3}\right)\Delta_{2}+\left(v_{3}-v_{2}\right)\Delta_{1}\right) \end{array}$$
(4)$$\begin{array}{cc} s & =\frac{d^{2}}{\Delta_{1}\Delta_{2}\Delta_{3}}\left(A\sin\theta +B\alpha \cos\theta \right) \end{array}$$
(5)$$\begin{array}{cc} t & =\frac{d^{2}}{\Delta_{1}\Delta_{2}\Delta_{3}}\left(B\alpha \sin\theta +A\cos\theta \right) \end{array}$$
where:(6)$$\begin{array}{cc} u & =\frac{d^{2}}{\Delta_{1}\Delta_{2}\Delta_{3}}u_{1} \end{array}$$
$$\begin{array}{cc} u_{1} & =\left(\left(v_{2}-v_{1}\right)\frac{\Delta_{3}}{2}+\left(v_{1}-v_{3}\right)\frac{\Delta_{2}}{2}+\left(v_{3}-v_{2}\right)\frac{\Delta_{1}}{2}+\left(u_{r1}-u_{r2}\right)v_{3}+\left(u_{r3}-u_{r1}\right)v_{2}+\left(u_{r2}-u_{r3}\right)v_{1}\right)\alpha d\\ \; & +\left(B\alpha \cos\theta +A\sin\theta \right)h\\ A & =\left(\left(u_{r1}-u_{0}\right)v_{2}+\left(u_{0}-u_{r2}\right)v_{1}+\left(u_{r2}-u_{r1}\right)v_{0}\right)\Delta_{3}+\left(\left(u_{0}-u_{r1}\right)v_{3}+\left(u_{r3}-u_{0}\right)v_{1}+\left(u_{r1}-u_{r3}\right)v_{0}\right)\Delta_{2}\\ \; & +\left(\left(u_{r2}-u_{0}\right)v_{3}+\left((u_{0}-u_{r3}\right)v_{2}+\left(u_{r3}-u_{r2}\right)v_{0}\right)\Delta_{1}\\ B & =\left(u_{r2}-u_{r1}\right)\Delta_{3}+\left(u_{r1}-u_{r3}\right)\Delta_{2}+\left(u_{r3}-u_{r2}\right)\Delta_{1} \end{array}$$

We define (ϵ) as an error in the pitch angle of the vision system. In the presence of such an error ϵIn the presence of an error (ϵ) in the pitch angle of the vision system, the planes with which the scene is reconstructed, called calculated planes, deviate from the ideal planes, which are the ones that are obtained when the error in the angle is within a certain amount. This deviation, which the authors have termed $\epsilon_{normal}$, is measured as the angle formed by the normal vectors to the two planes considered (calculated and ideal), that is:(7)$$\epsilon_{normal}=\arccos\left(\frac{\left|{\vec{n}}_{ideal}\cdot {\vec{n}}_{calculated}\right|}{\left|{\vec{n}}_{ideal}\right|\left|{\vec{n}}_{calculated}\right|}\right)$$

The study carried out in [1] allows us to affirm that the deviation between the calculated and ideal planes is limited by the value of ϵ, meaning the reconstruction technique does not introduce additional errors. In addition, it follows a beta distribution with α=2 and β=0.4, which means that in most cases the difference between the normals will have a value very close to the maximum, which, as just mentioned, is the value of ϵ. Ref. [1] also determines that the value of $\epsilon_{normal}$ depends only on the absolute value of the error (see Figure 2), and is invariant not only to the sign of ϵ but also to the distance to the image plane. The latter implies that the effect of an error in the pitch angle of the cameras on the reconstruction of the scene is the same regardless of the distance to the image plane at which the ideal plane is located.

Figure 2 shows the behavior of $\epsilon_{normal}$ versus ϵ for four different planes:(1)$\rho_{X}=0^{\circ };\rho_{Y}=45^{\circ };\rho_{Z}=45^{\circ }$(2)$\rho_{X}=15^{\circ };\rho_{Y}=75^{\circ };\rho_{Z}=0^{\circ }$(3)$\rho_{X}=30^{\circ };\rho_{Y}=15^{\circ };\rho_{Z}=90^{\circ }$(4)$\rho_{X}=45^{\circ };\rho_{Y}=75^{\circ };\rho_{Z}=0^{\circ }$

In the study presented in [1], only the value of the deviation between the normal of the ideal and calculated planes was analyzed. The axis around which this effect occurs was ignored. To make use of this tool as a calibration system, the deviation has to be fully characterized. To do this, we studied the behavior of this variable using the same set of planes described in [1].

The axis of rotation can be obtained as the vector product of the normal of the ideal and calculated planes and can be represented by its components on each axis $w_{x}$, $w_{y}$, and $w_{z}$. The different axes of rotation obtained for each value of ϵ for a specific ideal plane are shown in Figure 3. Figure 3 (right) shows a close-up view of the left image with a much smaller range on all three axes, providing a more detailed view. Note that the rotation axes are coplanar and that they are symmetrical around the value of $\epsilon =0^{\circ }$ within the plane that contains them. Since the axis of rotation is obtained as the vector product of the normals of the ideal and calculated planes, and the ideal plane for the whole observation is the same, this will be the plane that will contain all the axes of rotation.

The same information from the above figure can be shown superimposed with the deviation modulus between the normal vectors to the ideal and calculated planes. Thus, Figure 4 shows how the rotation axes are distributed in relation to the observed deviation. The *X* axis shows a combination of two parameters. The coordinates in the global reference system of each rotation axis have been shifted along the axis as many units as the ϵ value that generated it. For its part, the *Y* axis also combines the corresponding coordinate in the global coordinate system of the rotation axis with the angle of deviation between the normals. This provides a compact representation of both the information on the angle and the axis of rotation that is present between the normals to the ideal and calculated planes.

This analysis of the reconstruction system proposed allows us to ensure that any element in the real world can be represented using the technique described. In addition, the necessary tools are available to model the transformations that the ideal plane undergoes to become the calculated plane when it is projected in a vision system with a possible ϵ error in the pitch angle or θ angle.

All this provides us with the necessary tools to propose the calibration of a stereoscopic vision system like the one described in this paper by obtaining the ϵ value. The proposed strategy relies on the use of a machine learning tool that can estimate the value of ϵ from the coefficients of the ideal and calculated planes, as well as the value of the angle θ. Once this tool has been trained, the vision system will be presented with a series of images containing planes whose ideal coefficients are known. This way, once the calculated coefficients are obtained and since the θ angle is known, the trained system will be able to estimate the error in the pitch angle (ϵ).

### 2.2. Machine Learning

Machine Learning is a subfield of computer science and artificial intelligence that uses algorithms that allow machines to learn by imitating human behavior.

Among the different types of existing machine learning algorithms, those corresponding to supervised learning work with labeled data, that is, data for which the target answer is already known. Based on a data history set, it tries to look for patterns by relating them to a special field, called a target, given certain input variables.

With this data history, the algorithm can learn to assign an output label or function that allows it to predict the target attribute for a new action. Given the characteristics of the proposed calibration problem, these types of algorithms are best suited to our needs.

Supervised learning uses classification or regression algorithms, depending on the type of output data. Classification algorithms are used when the result is a discrete label. This means that they are used when the answer is based on a finite set of results. By contrast, regression analysis is the subfield whose goal is to establish a method for the relationship between a certain number of features and a continuous target variable.

Since the calibration system aims to estimate the value of ϵ as accurately as possible, with ϵ being a continuous variable, the problem was studied using different regressors. Specifically, we conducted a comparative study between:Linear Regression (LR)Regression Trees (RT)Regression Forest (RF)Multilayer Neural Networks (NN)

The tool selected to implement and study the regressors was MATLAB, MATLAB’s Statistics and Machine Learning Toolbox, and the Regression Learner App [23]. By selecting these tools, we were able to do a homogeneous analysis of the different regressors considered, and also set limits on the configurable parameters for each of them.

The following input features were considered for all of them, which will be described in detail in Section 2.4:Normal vector to the ideal planeVector normal to the calculated planeValue of angle θValue of angle $\epsilon_{normal}$Vector of the axis of rotation between normals

The output feature of the system is ϵ.

Due to the characteristics of the problem in question, which does not consider the change over time of the position of the vision system, but rather focuses on analyzing it in an instantaneous position, other potential approaches, like the one described in [24], which makes use of convolutional neural networks, were discarded. Since each input feature consists of a single data point, there is no benefit to doing convolutions in the different layers of the network, so this approach was not considered.

As described earlier, the axis of rotation of the deviation between the planes is a variable that can be calculated from the normal vectors to the ideal and calculated planes. However, given the nature of the selected regressors, the result of performing the regression with and without the information on the axes will be compared to assess the need for their use, which is why it is not included in the set of input characteristics that the regressors use.

### 2.3. Calibration Technique

The method proposed for calibrating the vision system using regressors consists of the following steps:Construction of a training set that is sufficiently representative of the variety of planes that can occur in a real scene, as well as the different orientations that the pitch angle (θ) and its error (ϵ) can have, and their effect in the reconstruction process.Training the selected regressor so that based on the data in the training set it can return the value of ϵ from the set of features described in Section 2.2. This process includes both the training stage and the validation of said training.Once the regressor is trained, a set of calibration planes whose normal vector is known is presented to the vision system. Since the pitch angle is also known, using the process described in Section 2.1 will yield the normal vector calculated for each of the planesThe set of calibration data (normal vectors to the ideal planes, normal vectors to the calculated planes and pitch angle) is input to the regressor so it can estimate the value of ϵ.For the calibration plane, an ϵ value is obtained that will be very similar, although probably not exactly equal because the regressor itself behaves differently depending on the characteristics of the input planes. Because of this, there is a final stage to determine the value of ϵ, which is done by taking the average value of the ϵ obtained for each calibration plane.

### 2.4. Input Data to the Regressors

As previewed in Section 2.2, a set of input features has been selected to train the regressors such that the relationship between the content of the scene and the status of the vision system can be adequately characterized.

Since the goal is to determine the value of the error (ϵ) of the pitch angle (θ), we need to know the transformation that occurs when reconstructing the three-dimensional information once it is captured by the vision system subject to the error.

The form chosen to represent this, based on the description given in Section 2.1, was to include in the features vector the normal vector to the ideal plane that is being considered, since it fully describes said plane. Similarly, the result of the transformation can be represented through the normal vector to the calculated plane. Finally, the value of the pitch angle (θ) of the vision system at the time when the plane is evaluated, together with the error (ϵ), completes the description of the features.

In addition to these features, it is possible to calculate the angle of rotation $\epsilon_{normal}$ between the normals to the planes and the axis around which the rotation occurs. As mentioned in Section 2.2, a comparative study will be carried out to determine if including these last features allows us to improve the regression results.

Taking into account that both the axis of rotation between the normals and the vector normal to the ideal plane and the vector normal to the calculated plane are three-dimensional variables, it follows that the features vector input to the regressor, consisting of both normal vectors and the value of θ, will have a dimension of 7 × 1 When the axis of rotation between the normals and the value of $\epsilon_{normal}$ are added to the features vector, its dimension becomes 11 × 1. The dimension of the output vector is 1 × 1, Figure 5 shows a graphic representation of the difference between the features vectors.

### 2.5. Description of the Regressors

Once the input and output features that the regressors will use have been presented, it is possible to describe in more detail each of the regressors that will be considered in this study:Linear Regression: Linear regression [25,26] is a statistical modeling technique used to describe a continuous response variable as a function of one or more predictor variables. These techniques are used to create a linear model. The variants considered calculating a series of coefficients that weigh the contribution of each input feature or set of features in estimating the output value.The Linear and Robust linear models approximate a linear function that does not combine the input variables. The difference between both models is in the type of linear regression used. While the Linear model uses simple linear regression, the Robust linear model makes use of robust linear regression.The Interactions linear and Stepwise linear models try to approximate an expression that, in addition to the input features, includes linear combinations between them, taken in pairs. In the case of Interactions linear, the training process considers the interaction effects among the inputs to achieve the best fit. Finally, the Stepwise linear training process adds or removes terms to the model.Regression Tree: Regression trees [27,28] are a nonparametric regression method that creates a binary tree by recursively splitting the data on the predictor values. The splits are selected so that the two child nodes have smaller variability around their average value than the parent node.Three models with fixed features and one that can be optimized have been used. In the models with fixed features, only the minimum leaf size varies, with 4, 12, and 36 being the values chosen for the Fine, Medium, and Coarse models, respectively. For its part, the Optimizable model, using Bayesian optimization, tries to determine the optimal minimum leaf size.Regression Forest: Regression forest [29] is an ensemble learning method for regression that works by constructing a multitude of decision trees at training time. An optimizable model has been chosen that automatically adjusts the following: the ensemble method, the minimum leaf size, the number of learners, the learning rate, and the number of predictors to be sampled.Multilayer Neural Network: Neural networks [30,31] are based on the functioning of the human brain. They are formed by different nodes that operate as neurons and transmit signals and information to each other. These networks receive different input information, process it as a whole, and generate an output with the predictions set according to how they have been programmed. In the study presented here, all the models chosen have an input layer with one neuron per input feature and an output layer with a single neuron, where the output is the estimated value. In all cases, ReLu has been used as the activation function. What differentiates the models is their architecture:**–** Narrow NN: A single hidden layer with 10 fully connected neurons.**–** Medium NN: A single hidden layer with 25 fully connected neurons.**–** Wide NN: A single hidden layer with 100 fully connected neurons.**–** Bilayered NN: Two hidden layers with 10 fully connected neurons each.**–** Trilayered NN: Three hidden layers with 10 fully connected neurons each.

## 3. Calculation

Having defined the tools selected to implement the calibration system proposed, this section details the procedure used to be able to use them.

### 3.1. Design of Experiments

The purpose of this work is to introduce the calibration technique proposed and demonstrate its validity to the problem posed. As a result, any reference to the processing of stereo image pairs, which can be consulted in [1], is ignored, and the analysis focuses on processing the information extracted from the process, which can be implemented in different ways. Since the starting point is the information obtained from processing the stereo pair, the authors have chosen to build a synthetic training set and parameterize it in such a way that it faithfully represents the variability of elements that can be found in a real scene.

Said training set is composed of multiple ideal planes in the global coordinate system represented by their parametric equation. The starting point is a plane parallel to the image plane. This plane is modified by rotation and translation transformations, generating new elements of the training set. Carrying out this process exhaustively, it is possible to generate a set of planes that sufficiently represent the possible elements of a real scene. These planes will be projected in the vision system described in this paper by varying the value of θ and ϵ to simulate the different scenarios to consider. These values have been limited to keep the training set from growing excessively. This was done by considering the most realistic scenarios based on a realistic structure of the vision system. Finally, the projected plane is reconstructed to yield the parametric equation of the corresponding calculated plane.

Following this procedure with the selected parameters, a training set of approximately 300,000 (295,323) input features vectors, with their respective outputs, has been constructed. As explained in [1], the error in the pitch angle was verified to be invariant to distance, so this variable was not taken into account when building the training set, since it would cause it to grow excessively without providing relevant information.

An analogous process was used for the test set, constructing a data set whose size was 25% of the validation set and containing data within the same ranges of values as those of the training set.

To select the regressors that best adapt to the problem, the training process was carried out in two stages. Firstly, all the regressors were trained using a holdout validation process, with 25% of the data for the validation set. Subsequently, the most promising regressors were re-trained with a 5-fold cross-validation process to obtain the best possible parameterization.

The metrics selected to determine which regressors offer the best performance were the Root Mean Square Error (RMSE), which has the property of being in the same units as the response variable, and Relative Absolute Error (RAE), which allows us to compare a mean error to errors produced by a trivial or naive model.
(8)$$\mathrm{RMSE}=\sqrt{\frac{\sum_{i=1}^{n}{\left(\epsilon_{calculated_{i}}-\epsilon_{ideal_{i}}\right)}^{2}}{n}}$$
(9)$$\mathrm{RAE}=\frac{\sum_{j=1}^{n}\left|\epsilon_{calculated_{j}}-\epsilon_{ideal_{j}}\right|}{\sum_{j=1}^{n}\left|\epsilon_{ideal_{j}}-\bar{\epsilon_{ideal}}\right|}$$

As is well known, lower values of RMSE indicate a better fit. In the same way, a good prediction model will present an RAE value close to zero. RMSE is a good measure of how accurately the model predicts the response, while RAE provides information in relative terms of how much the model deviates. Both accuracy and deviation are the most important criteria to adjust, since the primary purpose of the model is to predict.RMSE is a good measure of how accurately the model predicts the response and is the most important criterion to adjust since the primary purpose of the model is to predict.

### 3.2. Calibration Process

The results presented in the previous section can be used to select the regressor that is best suited to the problem considered herein. Once this tool is trained, the calibration process of the vision system can proceed. As discussed in Section 2.3, the calibration technique proposed requires a set of calibration planes whose ideal coefficients are known.

The set of calibration planes was chosen by selecting planes that are easily reproducible in a real scene, which facilitates the practical application of the technique described. The calibration process is carried out with more than one plane to avoid potential partial solutions.

Once the planes are selected, they are processed as described in Section 2.1 to obtain the calculated coefficients corresponding to the reconstruction for a known configuration of the vision system (θ and ϵ known). All these data are used to reconstruct the vector of input features that the regressor will process, returning as an output the estimated value of ϵ.

Finally, a consensus process is carried out among the ϵ values obtained for each calibration plane. The tool chosen for this phase is the mean.

The procedure presented in this paper for calibrating a stereoscopic vision system was repeated for multiple values of θ and ϵ. This allows us to affirm that not only does it behave as expected for a discrete set of cases, but that it can estimate the ϵ value within the error limits established earlier considering the most realistic scenarios, which correspond to values of ϵ and θ in the intervals $\left[-5^{\circ },5^{\circ }\right]$ in 0.25 degrees step and $\left[-10^{\circ },10^{\circ }\right]$ in 1-degree steps, respectively. This is an important result since it implies that this is a general-purpose calibration method.

## 4. Results

In the previous section, the methodology and experiments carried out to validate the proposed calibration method were presented. The purpose of this section is to analyze the results and show that they validate the hypothesis proposed. In addition, as mentioned above, the results obtained will be compared with the models with 7 and 11 input features.

### 4.1. Results of Experiments Carried Out with Different Regressors

Table 1 shows the results obtained for the set of regressors proposed in Section 3.2 without considering $\epsilon_{normal}$ and the rotation axis between normals and trained using a holdout validation, as explained earlier.

Table 2 shows the results obtained for the set of regressors proposed in Section 3.2 considering $\epsilon_{normal}$ and the rotation axis between normals and trained using a holdout validation, as explained earlier.

Comparing the results obtained in the cases shown in Table 1 and Table 2, it is observed that the simplest algorithms (linear regressors) significantly improve their results. However, in those regressors that already obtained good results, the performance does not experience a significant change. To try to obtain the best possible results, the entire set of regressors has been retrained considering the $\epsilon_{normal}$ value and the axis of rotation between the normals, using a 5-fold validation process. The results are shown in Table 3.

In light of the good results obtained with the tree-based regressors, and to obtain the best possible result, two optimizable implementations were used. As discussed in Section 2.5, this type of implementation allows certain regression parameters to be automatically adjusted to optimize their operation. Specifically, the Optimizable tree, which consists of a regression tree, and the Optimizable ensemble, which relies on a combination of different trees (ensemble), were used to form a Regression Forest (RF).

In the case of the Optimizable tree, the main parameter to be optimized is the minimum leaf size. The adjustment of this parameter is very similar to that of the fine tree, confirming the results of the fine, medium, and coarse trees.

For the Optimizable ensemble, in addition to the minimum leaf size parameter, there are other important ones such as the type of ensemble used, the number of learners, or the number of predictors to sample, which, once optimized, yield even better results than with a single regression tree.

The hyperparameters of each optimizable model resulting from the training and optimization process can be found at the bottom of Table 1, Table 2 and Table 3.

The graph in Figure 6 shows the discrepancy between the estimated and actual values of ϵ. Specifically, the estimated value of ϵ versus the actual value is shown for four regressors in Table 3. The degree of dispersion of the points with respect to the main diagonal is a measure of how good or bad the estimate is.

Figure 6 shows that there are cases in which the value estimated by the regressor deviates from the real value of ϵ to a greater extent than in most cases, meaning it has a much higher value of RMSE and/or RAE. This figure allows us to have a qualitative vision of how well the regressors behave, reaffirming the results presented in Table 3 These results were obtained considering the entire set of calibration planes, regardless of their configuration. In the second phase, the information is analyzed again in a disaggregated manner to discern whether it is possible to define some criterion for selecting the calibration planes that yield improved results.

To try to select the best set of calibration planes, defined as those that, once processed for different configurations of θ and ϵ, provide a lower RMSE and RAE, the RMSE information for each configuration of planes considered was segmented. Thus, in Figure 7 the RMSE values are represented as a function of the $\rho_{X}$, $\rho_{Y}$, and $\rho_{Z}$ rotations that gave rise to the plane considered. Since in a three-dimensional graph it is not possible to represent these four parameters ($\rho_{X}$, $\rho_{Y}$, $\rho_{Z}$, and RMSE), the information is shown in different graphs, corresponding to different values of $\rho_{Z}$.

Figure 7 shows that, for most configurations of the calibration plane, the RMSE value is lower than that given in Table 3. Analyzing the behavior also shows that for cases in which $\rho_{Z}=0^{\circ }$ and $\rho_{Z}=90^{\circ }$, the results are worse, meaning the regressor is not able to estimate the value of ϵ with the same precision as in the other cases. These planes are thus discarded for calibration. This filtering makes it possible to significantly reduce the RMSE and RAE values. Table 4 shows the RMSE and RAE values resulting from applying the filtering described to the regressors in Table 3. Therefore, to calibrate the vision system, we select a certain number of planes and estimate the value of ϵ as the average of the values calculated for each calibration plane used.

Figure 8 shows the effect of the filtering stage, where the red points represent the discarded planes. Thanks to this technique, most of the outliers are removed. This causes a significant improvement in the results, as can be seen in Table 4.

### 4.2. Discussion of Results

A number of results were presented throughout this paper to validate the calibration system proposed.

First, it was determined that the use of regressors, and in particular those derived from decision trees and certain neural network configurations, provide the best results. Although the problem information does not fall within the classification of categorical variables, where trees are known to yield good results, the problem has other characteristics that make it suitable to be approached using this type of algorithm. On the one hand, the information is not characterized by having a marked component of randomness, as could be the case with the pre-processing of the images. On the other, the dimensionality of the problem and the features used to solve it make it so that other regressors are unable to obtain similar results. This is the case of the poor fit obtained with the linear regressors. Similarly, the absence of a time component in the processing, since the images were considered independently and not as part of a sequence, means that other systems, such as convolutional networks, cannot leverage their benefits when solving this problem.

Secondly, the possible advantages of including the angle of rotation between plane normals in the set of input features to the regressors to estimate the value of ϵ have been analyzed. For this purpose, a comparison has been made of the results obtained with the different regressors after training them with and without these new features ($\epsilon_{normal}$ and the axis of rotation between the normals of the planes). Thus, it is found that the results obtained with the worst-performing regressors (linear regressors) improve significantly with the inclusion of the mentioned features. On the contrary, the results of the best-performing regressors hardly change.

A third result stems from analyzing the behavior of the selected regressors and specifically of the ones that provide the best results. It has been observed that the planes of the calibration set whose configurations contain an angle $\rho_{Z}=0^{\circ }$ or $90^{\circ }$ produce worse calibration results. This can be explained by analyzing how the planes of the different sets and, specifically, of the calibration set are constructed. The first rotation is done about the *Z* axis. As a consequence, a rotation of 0 or 90 degrees will produce changes to the plane that are not easily distinguishable in the regression stage. For this reason, the RMSE values obtained are worse in this case. This is why Figure 6 shows multiple classification errors, around the value 0, since the regressor is not able to correctly identify the value of ϵ.

## 5. Conclussions

In this paper, the authors present a novel method for calibrating the pitch angle in a stereoscopic vision system. They do so by using techniques based on U-V disparity to interpret the three-dimensional information contained in a scene as a set of planes. This provides a tool that can be used to estimate the error in the reconstruction of a certain element in the scene (a plane) caused by a certain deviation (ϵ) in the pitch angle.

To calibrate the vision system, the authors propose using the knowledge generated to train a regressor. The value of ϵ is then estimated using the information from the reconstructed set of planes. Successive reconstructions can use this value to compensate for this error.

Although the proposed technique makes use of U-V disparity, the calibration system is completely independent of the visualization techniques to be used with the calibrated system. This is an important feature since the aim of the authors is to provide a generic tool for correcting an intrinsic error in the stereo vision system.

To provide a regressor that can adapt to the situations that may arise in a real-world scene, it has been trained using a training set containing on the order of 300,000 elements. This makes it possible to have an adequate representation of the different orientations that real-world elements may exhibit. The different regressors were thus trained, with certain configuration of neural networks, regression trees, and regression forests yielding the best results.

Although a priori the types of features used to train the regressors are not ideally suited to the regressors that provided best results, since they are not categorical variables, the absence of randomness in the nature of the information and the use of independent images meant that these types of regressors provided the best results.

By studying in detail the set of training planes, we determined that those that were generated by using the $\rho_{Z}$ rotation with a value of $0^{\circ }$ or $90^{\circ }$ introduce greater uncertainty, increasing both the RMSE and the RAE of the regressor. Because of this, the planes that satisfied any of these premises of the calibration set were excluded, thus improving the system response.

In all, the results presented demonstrate that the system described and implemented can correctly estimate the potential error present in the pitch angle with a very low RMSE and RAE. This correction can then be introduced in any subsequent processing of the information extracted from the images.

## Figures and Tables

**Figure 1 sensors-23-00212-f001:**
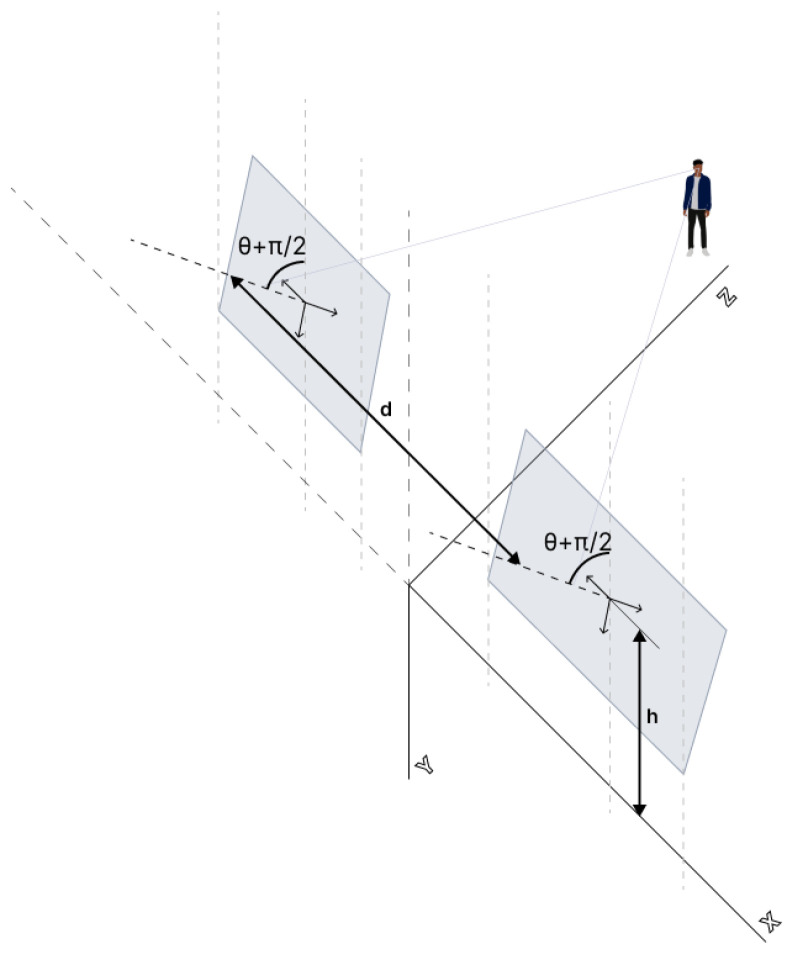
Schematic representation of a stereo vision system.

**Figure 2 sensors-23-00212-f002:**
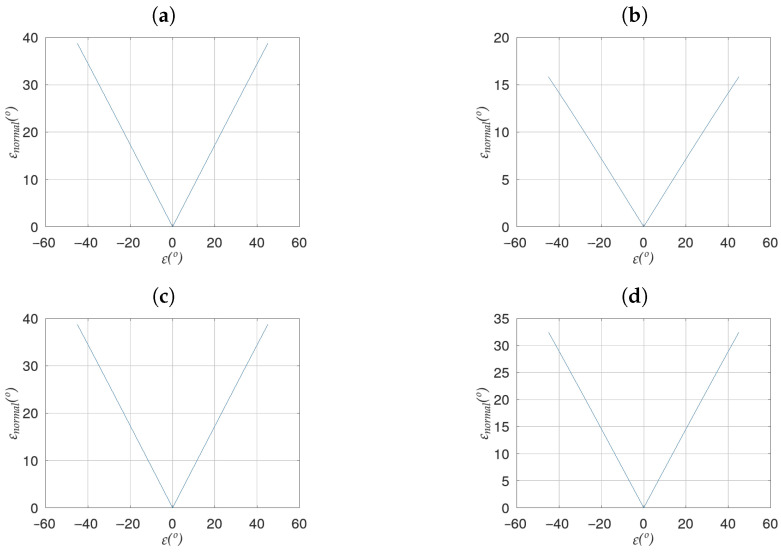
Deviation between the normals of the ideal and calculated planes as a function of the error in angle θ for: (**a**) Plane 1, (**b**) Plane 2, (**c**) Plane 3, (**d**) Plane 4.

**Figure 3 sensors-23-00212-f003:**
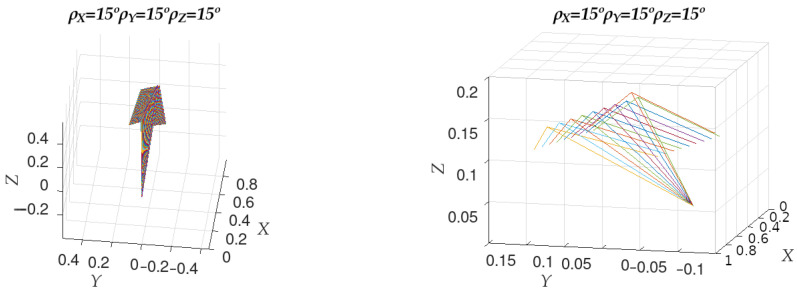
On the left, the orientation of the axes of rotation between the normals of the ideal and calculated planes. On the right, a close-up of the figure on the left.

**Figure 4 sensors-23-00212-f004:**
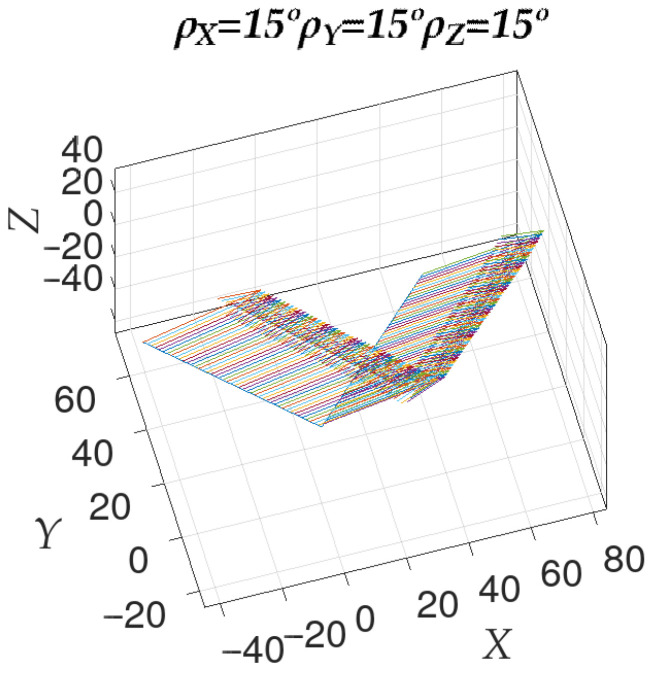
Orientation of the axes of rotation between the normals of the ideal and calculated planes, superimposed with the value of ϵ that gave rise to that axis of rotation.

**Figure 5 sensors-23-00212-f005:**
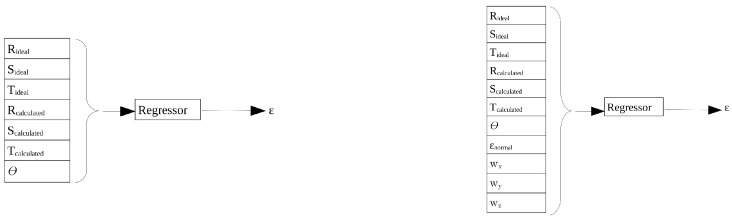
Schematic representation of the information that the regressor receives as input and returns as output (estimated value of ϵ). On the left, the inputs are the coefficients of the ideal and calculated planes and θ value. On the right, four new features have been added as inputs: the coefficients of the axis of rotation between normals and $\epsilon_{normal}$ value.

**Figure 6 sensors-23-00212-f006:**
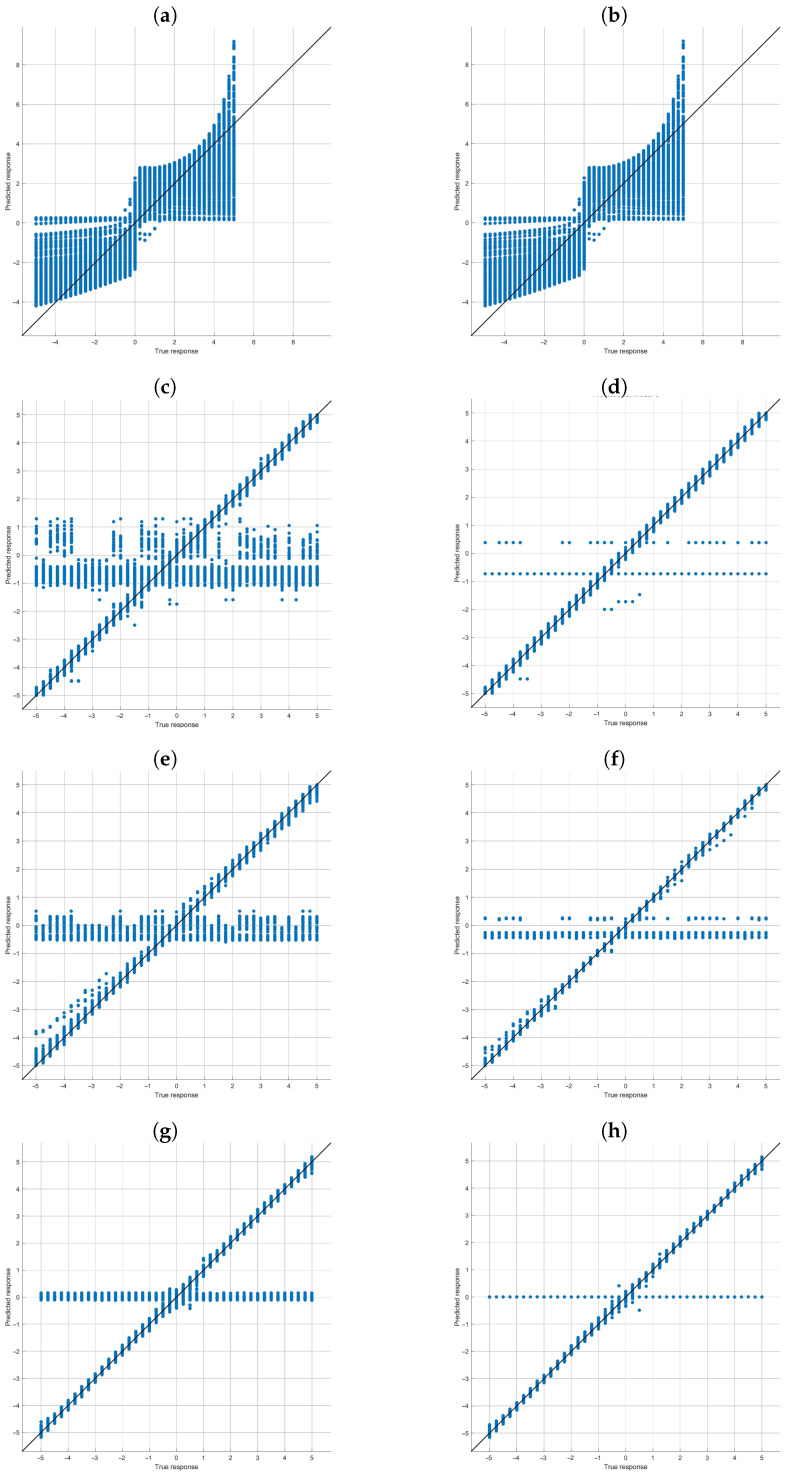
Comparison between the predicted response (x-axis) and the real value (y-axis) for four of the regressors of Table 3 (**a**) Linear regression validation, (**b**) Linear regression test, (**c**) Optimizable tree validation, (**d**) Optimizable tree test, (**e**) Optimizable ensemble validation, (**f**) Optimizable ensemble test, (**g**) Wide neural network validation and (**h**) Wide neural network test.

**Figure 7 sensors-23-00212-f007:**
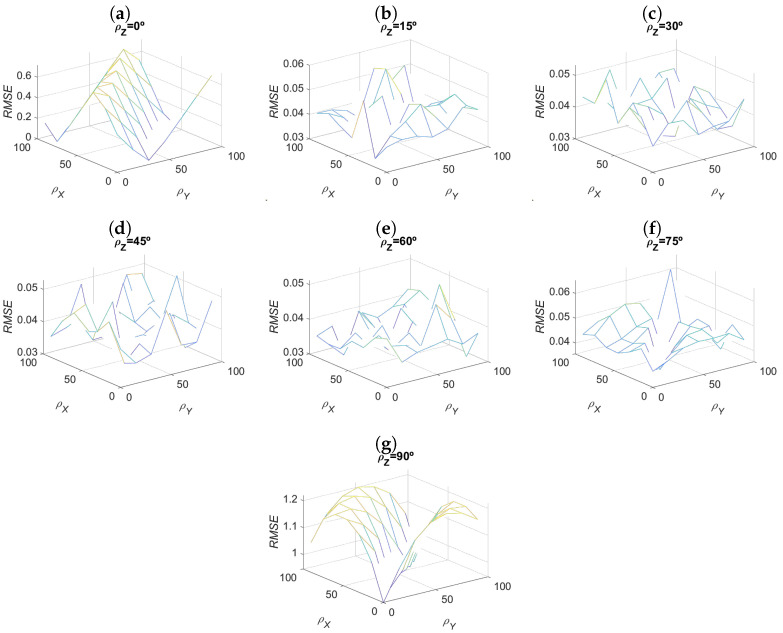
RMSE calculated for each calibration plane configuration of wide neural network regressor. Given the number of variables to represent, it was necessary to separate the representation into multiple graphs for the different values of $\rho_{Z}$: (**a**) $\rho_{Z}=0^{\circ }$, (**b**) $\rho_{Z}=15^{\circ }$, (**c**) $\rho_{Z}=30^{\circ }$, (**d**) $\rho_{Z}=45^{\circ }$, (**e**) $\rho_{Z}=60^{\circ }$, (**f**) $\rho_{Z}=75^{\circ }$ and (**g**) $\rho_{Z}=90^{\circ }$.

**Figure 8 sensors-23-00212-f008:**
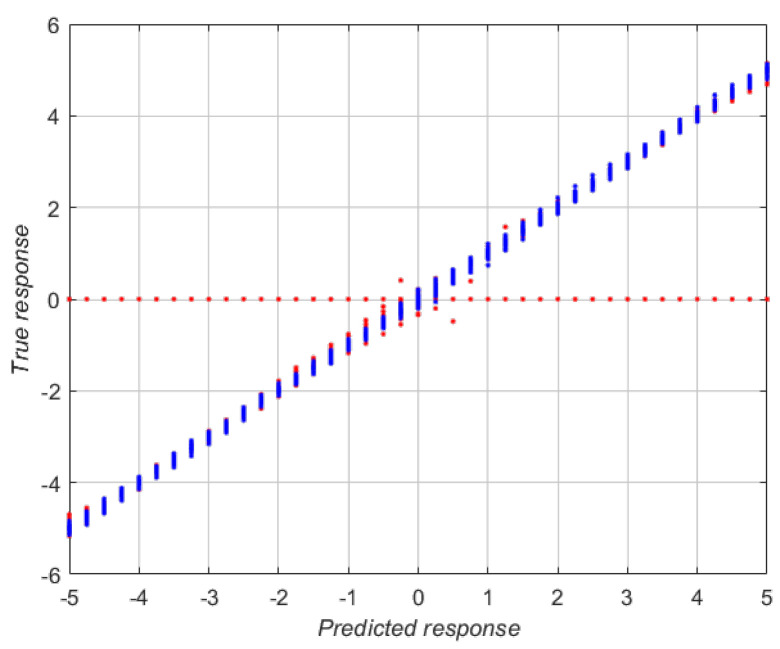
Comparison between the predicted response (x-axis) and the real value (y-axis) of the wide neural network after filtering the input planes. Blue points represent the planes that pass the filtering stage meanwhile red points represent the discarded ones.

**Table 1 sensors-23-00212-t001:** Results of training the regressors without considering $\epsilon_{normal}$ and the rotation axis between normals with holdout validation.

Regressor	Variant	RMSE (${}^{\circ }$) Validation	RMSE (${}^{\circ }$) Test	RAE (%) Validation	RAE (%) Test
LR	Linear	2.2470	2.2491	70.66	70.72
LR	Interactions	1.2526	1.2600	34.59	34.65
LR	Robust	2.5528	2.5501	68.70	68.85
LR	Stepwise	1.2525	1.2600	34.59	34.65
RT	Fine	0.4801	0.4870	4.11	4.26
RT	Medium	0.4797	0.4870	4.11	4.26
RT	Coarse	0.4851	0.4887	4.33	4.49
RT	Optimizable ^1^	0.4797	0.4870	4.11	4.26
RF	Optimizable ensemble ^2^	0.4707	0.4804	3.57	3.72
NN	Narrow	0.5684	0.6177	13.43	13.53
NN	Medium	0.4845	0.4930	6.98	7.09
NN	Wide	0.4569	0.4689	3.63	3.75
NN	Bilayered	0.4719	0.4841	6.01	6.15
NN	Trilayered	0.4729	0.4757	5.07	5.19

^1^ Hyperparameters optimized: Minimum leaf size = 14. ^2^ Hyperparameters optimized: Ensemble method = Bag, Number of learners = 10, Minimum leaf size = 1 and Number of predictors to sample = 3.

**Table 2 sensors-23-00212-t002:** Results of training the regressors considering $\epsilon_{normal}$ and the rotation axis between normals with holdout validation.

Regressor	Variant	RMSE (${}^{\circ }$) Validation	RMSE (${}^{\circ }$) Test	RAE (%) Validation	RAE (%) Test
LR	Linear	1.4455	1.4479	45.36	45.31
LR	Interactions	0.6252	0.6368	12.59	12.68
LR	Robust	1.4479	1.4510	45.07	45.03
LR	Stepwise	0.6251	0.6368	12.59	12.68
RT	Fine	0.4707	0.4850	3.33	3.49
RT	Medium	0.4705	0.4850	3.33	3.49
RT	Coarse	0.4716	0.4858	3.44	3.60
RT	Optimizable ^1^	0.4706	0.4858	3.44	3.60
RF	Optimizable ensemble ^2^	0.4665	0.4838	4.08	4.23
NN	Narrow	0.4787	0.5035	7.45	7.57
NN	Medium	0.4587	0.4785	5.16	5.28
NN	Wide	0.4502	0.4692	3.69	3.81
NN	Bilayered	0.4571	0.4799	5.35	5.46
NN	Trilayered	0.4755	0.4795	5.35	5.47

^1^ Hyperparameters optimized: Minimum leaf size = 13. ^2^ Hyperparameters optimized: Ensemble method = LBoost, Number of learners = 494, Minimum leaf size = 12,421 and Number of predictors to sample = 11.

**Table 3 sensors-23-00212-t003:** Results of training the regressors considering $\epsilon_{normal}$ and the rotation axis between normals with 5-fold validation.

Regressor	Variant	RMSE (${}^{\circ }$) Validation	RMSE (${}^{\circ }$) Test	RAE (%) Validation	RAE (%) Test
LR	Linear	1.4458	1.4479	45.36	45.31
LR	Interactions	0.6278	0.6368	12.59	12.68
LR	Robust	1.4484	1.4510	45.07	45.03
LR	Stepwise	0.6280	0.6368	12.59	12.68
RT	Fine	0.4711	0.4850	3.33	3.49
RT	Medium	0.4705	0.4850	3.33	3.49
RT	Coarse	0.4711	0.4858	3.44	3.60
RT	Optimizable ^1^	0.4705	0.4850	3.33	3.49
RF	Optimizable ensemble ^2^	0.4571	0.4715	3.39	3.52
NN	Narrow	0.4899	0.5173	8.39	8.50
NN	Medium	0.4635	0.4790	5.24	5.36
NN	Wide	0.4544	0.4690	3.64	3.75
NN	Bilayered	0.4650	0.4809	5.59	5.71
NN	Trilayered	0.4613	0.4778	5.11	5.24

^1^ Hyperparameters optimized: Minimum leaf size = 16. ^2^ Hyperparameters optimized: Ensemble method = Bag, Number of learners = 10, Minimum leaf size = 1 and Number of predictors to sample = 2.

**Table 4 sensors-23-00212-t004:** Results of filtering the planes used for calibration with regressors of Table 3.

Regressor	Variant	RMSE (${}^{\circ }$) Validation	RMSE (${}^{\circ }$) Test	RAE (%) Validation	RAE (%) Test
LR	Linear	1.3479	1.3429	42.96	42.78
LR	Interactions	0.4172	0.4149	9.88	9.85
LR	Robust	1.3491	1.3448	42.61	42.46
LR	Stepwise	0.4172	0.4149	9.88	9.85
RT	Fine	0.0523	0.0521	0.88	0.88
RT	Medium	0.0523	0.0521	0.88	0.88
RT	Coarse	0.0579	0.0578	0.98	0.98
RT	Optimizable	0.0523	0.0521	0.88	0.88
RF	Optimizable ensemble	0.0367	0.0366	0.99	0.99
NN	Narrow	0.2101	0.2093	5.94	5.93
NN	Medium	0.0959	0.0958	2.75	2.75
NN	Wide	0.0432	0.0432	1.28	1.27
NN	Bilayered	0.1092	0.1091	3.20	3.20
NN	Trilayered	0.0942	0.0943	2.67	2.68

## Data Availability

Not applicable.

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
