# Peer review of "Calibration of a Stereoscopic Vision System in the Presence of Errors in Pitch Angle"

_sensors, 2022, doi:10.3390/s23010212_

Round 1

Reviewer 1 Report

The manuscript is of interest for the application of stereoscopic sensors in areas such as robot control or autonomous driving. In these areas, automatic self-calibration is useful, since other techniques require out-of-service calibration with defined objects. In contrast, in the context of photogrammetric measurement systems, I cannot see any advantage of the presented method.

From my point of view, the benefit of the presented system compared to classical calibration techniques should therefore already be emphasized more clearly in the introduction. 

In Fig 1, the height of the camera above the ground plane is noted as "a", whereas in the related text it is called "h". In addition, I would suggest enlarging Fig. 1 and perhaps also change perspective to make the details (esp. the angle) clearer. This figure in its present form is almost identical (except for the designation "a") to Figure 1 in the authors' previous publication "Influence of Pitch Angle Errors in 3D Scene Reconstruction Based on U-V Disparity: A Sensitivity Study", cited in the text (Reference 1).

I would also suggest enlarging Fig. 3 in order to make single lines of the diagram visible.

The statement that “those regions of space in which the gradient of the disparity is constant represent the projection of points that are part of the same object” (Page 4, unnumbered Text lines between formula 1 and 2) should be further explained.

At the end of the same paragraph, the “y” should be replaced by “and” in the listing of formulas.

Reviewer 2 Report

Please address the following (13) concerns:

1. Regarding the selection of ML algorithms, the authors should provide a more precise intuition as to why they selected them over others.

2. The authors should at least provide a brief yet mathematical description of the algorithms they used.

3. The authors should provide information regarding the hyperparameter settings given to their ML algorithms, specifically on the best ML algorithm they evaluated. Such an approach can provide better reproducibility and validity of results.

4. The authors should provide more references, specifically in their methods, as some of their methods are standard. In addition, the references should suffice most of the claims made in the paper to strengthen its novelty. Adding references can also help alleviate certain ambiguities found in the paper.

5. The authors should provide a comprehensive table highlighting the strengths and weaknesses of previous methods on the identified problem. From there, the author should point out and highlight the formidability of their paper.

6. Can the authors provide an application-level or visual representation or results of how well their solution worked in a realistic scenario?

7. The authors must provide more details and evidence regarding their claim that “Although the authors base their method on U-V disparity and employ this same technique to completely reconstruct the 3D scene, one of the most interesting features of the method proposed is that it can be applied regardless of the technique used to carry out said reconstruction.

8. The authors should provide enough information on how they calculated their results.

9. The authors should provide more detailed information about their dataset and how they handled it.

10. Can the authors provide a comparison of this solution with existing state-of-the-art methods? If possible, can the authors also provide the pros and cons of their solution against those?

11. Can the authors provide more details about the said optimizable variant?

12. Writing requires a significant amount of attention and action.

Overall, the paper is quite interesting, but the authors require significant work and must provide adequate actions before any further decision. 

Reviewer 3 Report

Reading this paper, my first problem was that the authors assumed that the readers read [1] on which their paper is based. This of course is not usually true and the paper should be in general self-contained.

Here are some examples of problems that I noticed due to this assumption.

1)     What do the authors mean by pitch angle theta mentioned in the first paragraph?

2)     What do r,s,t,u mean in equation 2-5.

3)     How is the error epsilon defined? Error in pitch angle, error in the stereo system between the cameras or anything else. The error has to be defined accurately.

This all has to be fixed to make the paper self-contained.

2) It's not clear how realistic the setting is. The authors assume that besides eps everything is accurate. In real systems there is always some noise in the calculated normal. How does this affect the results?

3) In the experiments the simulated error is between -40 +40 degrees. These are very high numbers. Why would you assume such large errors? I would concentrate on much smaller errors. What also is important to measure is the relative error not the error itself. If for small errors (1 degree) the error in the estimation is 0.5 degree that is a problem.

4) The input to the classifier is the two normals and theta. You could also add the axis of rotation and eps_normal. It's true that they can be calculated from the given parameters but I don't think the classifiers are able to do that. In any case adding additional features which could be calculated from the other features is not problematic and might improve the results. You can compare the results with and without these additional features.

5) Its strange that no real experiments were conducted on real images. This at least could be used to explain the design of the experiments.

6) There is a whole field of multiple view geometry which deals among other things with stereo. It seems that the authors are not aware of the thousands of papers written on the subject. The authors should cite some of these papers. The textbook on the subject was written by Hartley & Zissermann. Using this knowledge,  for example given eps eps_normal could be calculated.

Round 2

Reviewer 2 Report

The authors have addressed my concerns adequately.

Reviewer 3 Report

My comments were addressed